# Towards biologically plausible Dreaming and Planning in recurrent spiking networks

## Abstract

Humans and animals can learn new skills after practicing for a few hours, while current reinforcement learning algorithms require a large amount of data to achieve good performances. Recent model-based approaches show promising results by reducing the number of necessary interactions with the environment to learn a desirable policy. However, these methods require biological implausible ingredients, such as the detailed storage of older experiences, and long periods of offline learning. The optimal way to learn and exploit word-models is still an open question. Taking inspiration from biology, we suggest that dreaming might be an efficient expedient to use an inner model. We propose a two-module (agent and model) **spiking** neural network in which "dreaming" (living new experiences in a model-based simulated environment) significantly boosts learning. We also explore "planning", an online alternative to dreaming, that shows comparable performances. Importantly, our model does not require the detailed storage of experiences, and learns online the world-model and the policy. **Moreover, we stress that our network is composed of spiking neurons, furhter increasing the biological plausibility and implementability in neuromorphic hardware.**

## 1 Introduction

Humans can learn a new ability after practicing a few hours (e.g., driving or playing a game), while to solve the same task artificial neural networks require millions of reinforcement learning trials in virtual environments. And even then, their performances might be not comparable to human's ability. Humans and animals, have developed an understanding of the world that allow them to optimize learning. This relies on the building of an inner model of the world. Model-based reinforcement learning Ye et al. (2021); Abbeel et al. (2006); Schrittwieser et al. (2020); Ha & Schmidhuber (2018); Kaiser et al. (2019); Hafner et al. (2020) have shown to reduce the amount of data required for learning. However, these approaches do not provide insights on biological intelligence since they require biologically implausible ingredients (storing detailed information of experiences to train models, long off-line learning periods, expensive Monte Carlo three search to correct the policy). Moreover, the storage of long sequences is highly problematic on neuromorphic and FPGA platforms, where memory resources are scarce, and the use of an external memory would imply large latencies. The optimal way to learn and exploit the inner-model of the world is still an open question. **Taking inspiration from biology, we explore an intriguing idea that a learned model can be used when the neural network is offline.** In particular, during deep-sleep, dreaming, and day-dreaming.

Sleep is known to be essential for awake performances, but the mechanisms underlying its cognitive functions are still to be clarified. A few computational models have started to investigate the interaction between sleep (both REM and NREM) and plasticity González-Rueda et al. (2018); Wei et al. (2016; 2018); Korcsak-Gorzo et al. (2020); Golosio et al. (2021) showing improved performances, and reorganized memories, in the after sleep network. **A wake-sleep learning algorithm has shown the possibility to extend the acquired knowledge with new symbolic abstractions and to train the neural network on imagined and replayed problems Ellis et al. (2020).** However, a clear and coherent understanding of the mechanisms that induce generalized beneficial effects is missing. The idea that dreams might be useful to refine learned skill is fascinating and requires to be explored experimentally and in theoretical and computational models.

Here, we define "dreaming" as a learning phase of our model, in which it exploits offline the inner-model learned while "awake". During "dreaming" the world-model replaces the actual world, to allow an agent to live new experiences and refine the behavior learned during awake periods, even when the world is not available. We show that this procedure significantly boosts learning speed. This choice is a biologically plausible alternative to experience replay Wang et al. (2016); Munos et al. (2016), which requires storing detailed information of temporal sequences of previous experiences. **Indeed, even though the brain is capable to store episodic memories, it is unlikely that it is capable to sample offline from ten or hundreds of past experiences.**

Usually, dreaming approaches are inefficient because of the large compounding errors associated to simulate long sequences in model-based simulated environment. In Ha & Schmidhuber (2018); Hafner et al. (2020); Okada & Taniguchi (2021), the authors have been able to train a neural network only in its dreams. In Ha & Schmidhuber (2018) the authors used evolutionary methods to optimize the policy, that are not a biologically plausible option, if compared to the reward based policy gradient. In Hafner et al. (2020) the authors simulated short sequences ($T = 15$), from initial condition belonging to past experiences. We show that we are able to successfully train the agent in long simulated sequences ($T = 50$), starting from random initial conditions.Also, we defined "planning", an online alternative to "dreaming", that allows to reduce the compounding-error by simulating online shorter sequences. However, this requires additional computational resources while the network is already performing the task. As stated above, learning the world-model usually requires the storage of an agent's experiences Ye et al. (2021); Abbeel et al. (2006); Schrittwieser et al. (2020); Ha & Schmidhuber (2018); Kaiser et al. (2019); Hafner et al. (2020), to offline learn the model. We circumvent this problem by formulating an efficient learning rule, local in space and time, that allows learning online the world-model. In this way, no information storage (except the network parameters) is required.

For the sake of biological plausibility, we consider the most prominent model for neurons in the brain: leaky integrate-and-fire neurons. But it is an open problem how recurrent networks of spiking neurons (RSNNs) can learn Bellec et al. (2020); Muratore et al. (2021); Capone et al. (2022), i.e., how their synaptic weights can be modified by local rules for synaptic plasticity so that the computational performance of the network improves. Despite recent advances in this direction Bellec et al. (2020); Göltz et al. (2021); Kheradpisheh & Masquelier (2020), there are yet very few applications to relevant tasks due to the difficulty to analytically tackle the discontinuity of the nature of the spike.

To our knowledge, there are no previous works proposing biologically plausible model-based reinforcement learning in recurrent spiking networks. Our work is a step toward building efficient neuromorphic systems **where memory and computation are integrated in the same support, network of neurons. This allows to avoid the problem of latency when accessing to external storages, e.g. to perform experience replay**. **Indeed, we argue that the use of (1) spiking neurons, (2) online learning rules, (3) memories stored in the network (and not in an external storage), make our model, almost straightforward to be efficiently implemented in a neuromorphic hardware.**

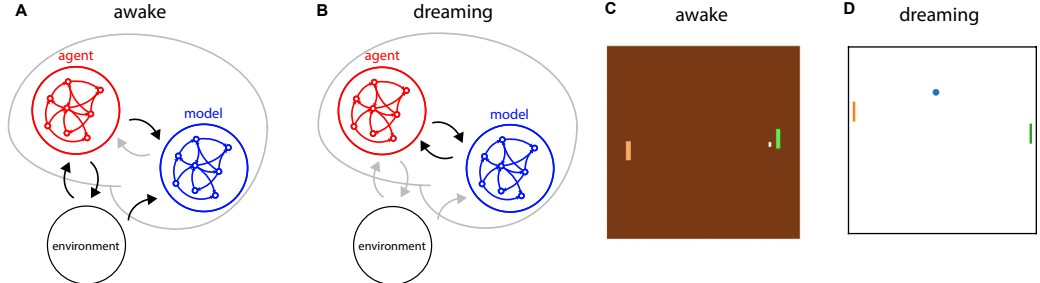

Figure 1: The agent-model spiking network. (A-B) Network structure: the network is composed of two modules: the agent and the model sub-networks. During the "awake" phase the two networks interact with the environment, learning from it. During the dreaming phase the environment is not accessible, and the model replaces the role of the environment. (C) Example frame of the environment perceived by the network. (D) Example of the reconstructed environment during the dreaming phase.

## 2 METHODS

We propose a two-modules recurrent spiking network (See Fig.1A), one computing the policy to behave in an environment, and the other learning to predict the next state of the environment, given the past states, and the action selected by the agent.

### 2.1 THE SPIKING NETWORK

Each module $\alpha$ ($\alpha = A$ for the agent network and, $\alpha = M$ for the model one) is composed of $N = 500$ neurons. Neurons are described as real-valued variable $v_{\alpha,i}^t \in \mathbb{R}$, where the $i \in \{1, \ldots, N\}$ label identifies the neuron and $t \in \{1, \ldots, T\}$ is a discrete time variable. Each neuron exposes an observable state $s_{\alpha,i}^t \in \{0, 1\}$, which represents the occurrence of a spike from neuron $i$ of the module $\alpha$ at time $t$. We then define the following dynamics for our model:

$$
\begin{cases}
\hat{s}_{\alpha,i}^t = \exp\left(-\frac{\Delta t}{\tau_s}\right) \hat{s}_{\alpha,i}^{t-1} + \left(1 - \exp\left(-\frac{\Delta t}{\tau_s}\right)\right) s_{\alpha,i}^t \\
v_{\alpha,i}^t = \exp\left(-\frac{\Delta t}{\tau_m}\right) v_{\alpha,i}^{t-1} + \left(1 - \exp\left(-\frac{\Delta t}{\tau_m}\right)\right) \left(\sum_j w_{ij}^\alpha \hat{s}_{\alpha,j}^{t-1} + I_{\alpha,i}^t + v_{\text{rest}}\right) - w_{\text{res}} s_{\alpha,i}^{t-1} \\
s_{\alpha,i}^{t+1} = \Theta\left[v_{\alpha,i}^t - v^{\text{th}}\right]
\end{cases}
\tag{1}
$$

$\Delta t$ is the discrete time-integration step, while $\tau_s$ and $\tau_m$ are respectively the spike-filtering time constant and the temporal membrane constant. Each neuron is a leaky integrator with a recurrent filtered input obtained via a synaptic matrix $w_{ij}$ (from neuron $j$ to neuron $i$) and an external input current $I_{\alpha,i}^t$. $w_{\text{res}} = -20$ accounts for the reset of the membrane potential after the emission of a spike. $v^{\text{th}} = 0$ and $v_{\text{rest}} = -4$ are the threshold and the rest membrane potential. The input to the agent network is a random projection of the state describing the world $I_{A,i}^t = \sum_h w_{ih}^{A,in} \xi_h^t$ (where $w_{ih}^{A,in}$ are randomly extracted from a Gaussian distribution with zero mean and variance $\sigma^{A,in} = 5$). The input to the model network is a projection of the action chosen by the agent and of variables describing the world state $I_{M,i}^t = \sum_h w_{ih}^{M,in,a} 1_{a^t=h} + \sum_h w_{ih}^{M,in,\xi} \xi_h^t$ ($w_{ih}^{M,in,a}$ and $w_{ih}^{M,in,\xi}$ are randomly extracted from a Gaussian distribution with zero mean and variance respectively $\sigma^{M,in,a} = 5$ and $\sigma^{M,in,\xi} = 5$), where we used the one-hot encoded action $1_{a^t=k}$ at time t, which assumes the value 1 if and only if $a^t = k$ (else it has value 0).

Finally, we define the following quantities, that are relevant for the learning rules described in the following sections. $p_{\alpha,i}^t$ is the pseudo-derivative (similarly to Bellec et al. (2020); Capone et al. (2022)) $p_i^t = \frac{e^{v_i^t/\delta v}}{\delta v (e^{v_i^t/\delta v}+1)^2}$ (it peaks at $v_i^t = 0$ and $\delta v$ defines its width) and $e_{\alpha,j}^t = \frac{\partial v_{\alpha,i}^t}{\partial w_{ij}^\alpha}$ the spike response function that can be computed iteratively as

$$
e_{\alpha,j}^{t+1} = \exp\left(-\frac{\Delta t}{\tau_m}\right) e_{\alpha,j}^t + \left(1 - \exp\left(-\frac{\Delta t}{\tau_m}\right)\right) \hat{s}_{\alpha,i}^t.
\tag{2}
$$

### 2.2 LEARNING THE WORLD MODEL

We train the model-network to predict the following state of the world $\xi_i^{t+1}$ and reward $r^t$ given the current state $\xi_i^t$ and the agent action $a^t$. The state $\xi_i^{t+1}$ and reward $r^t$ are encoded as linear readouts $\xi_k^t = \sum_i R_{ki}^\xi \bar{s}_{M,i}^t$, $r^t = \sum_i R_i^r \bar{s}_{M,i}^t$ of the spiking activity of the network, where $\bar{s}_{\alpha,i}^t$ is a temporal filtering of the spikes $s_{\alpha,i}^t$, where $\Delta t$ is the temporal bin of the simulation and $\tau_\star$ the timescale of the filtering:

$$
\bar{s}_{\alpha,i}^t = \exp\left(-\frac{\Delta t}{\tau_\star}\right) \bar{s}_{\alpha,i}^{t-1} + \left(1 - \exp\left(-\frac{\Delta t}{\tau_\star}\right)\right) s_{\alpha,i}^t
\tag{3}
$$

In order to train the network to reproduce at each time the desired output vector $\{\xi_k^{\star t}, r^{\star t}\}$, it is necessary to minimize the loss function:

$$
E^M = c_\xi \sum_{t,k} \left(\xi_k^{\star t} - \xi_k^t\right)^2 + c_r \sum_t \left(r^{\star t} - r^t\right)^2.
\tag{4}
$$

where $c_\xi = 1.0$ and $c_r = 0.1$ are arbitrary coefficient. It is possible to derive the learning rules by differentiating the previous error function (similarly to what was done in Bellec et al. (2020); Capone et al. (2022)):

$$
\begin{cases}
\Delta w_{ij}^M \propto \frac{dE}{dw_{ij}} \simeq \sum_t \left[ c_\xi \sum_k \mathsf{R}_{ik}^\xi \left( \xi_k^{\star t+1} - \xi_k^{t+1} \right) + c_r \mathsf{R}_i^r \left( r^{\star t+1} - r^{t+1} \right) \right] p_{M,i}^t e_{M,j}^t \\
\Delta \mathsf{R}_{ik}^\xi \propto \frac{dE^M}{d\mathsf{R}_{ik}^\xi} = \sum_t c_\xi \left( \xi_k^{\star t+1} - \xi_k^{t+1} \right) \bar{s}_{M,k}^t \\
\Delta \mathsf{R}_k^r \propto \frac{dE^M}{d\mathsf{R}_k^r} = \sum_t c_r \left( r^{\star t+1} - r^{t+1} \right) \bar{s}_{M,k}^t
\end{cases}
\tag{5}
$$

The resulting learning rule is local both in space (the synaptic update only depends on the presynaptic and postsynaptic neurons) and in time (the update at time $t$ does not depend on future times). This allows to train online the recurrent spiking model-network.

## 2.3 LEARNING THE AGENT'S POLICY

The policy $\pi_k^t$ defines the probability to pick the action $a^t = k$ at time $t$, and it is defined as:

$$
\pi_k^t = \frac{\exp\left( y_k^t \right)}{\sum_i \exp\left( y_k^t \right)}
\tag{6}
$$

where, $y_k^t = \sum_i R_{ki}^\pi \bar{s}_{\alpha,i}^t$. Following a policy gradient approach to maximize the total reward obtained during the episode, it is possible to write the following loss function (see Bellec et al. (2020); Sutton & Barto (2018)):

$$
E^A = - \sum_t R^t log(\pi_k^t)
\tag{7}
$$

where $R^t = \sum_{t' \geq t} r^{t'} \gamma^{t'-t}$ is the total future reward (or return), and $\gamma = 0.99$ is the discount factor. Using the one-hot encoded action $1_{a^t=k}$ at time t, which assumes the value 1 if and only if $a_t = k$ (else it has value 0), we arrive at the following synaptic plasticity rules (the derivation is the same as in Bellec et al. (2020), **to which we refer** for a complete derivation):

$$
\begin{cases}
\Delta w_{ij}^A \propto \frac{dE^A}{dw_{ij}^A} \simeq \sum_k r^t R_{ik}^\pi \sum_{t' \leq t} \gamma^{t-t'} \left( 1_{a^{t'}=k} - \pi_k^{t'} \right) p_{A,i}^{t'} e_{A,j}^{t'} \\
\Delta R_{ik}^\pi \propto \frac{dE^A}{dR_{ik}^\pi} = \sum_k r^t \sum_{t' \leq t} \gamma^{t-t'} \left( 1_{a^{t'}=k} - \pi_k^{t'} \right) \bar{s}_{A,i}^{t'}
\end{cases}
\tag{8}
$$

Intuitively, given a trial with high rewards, policy gradient changes the network output $y_k^t$ to increase the probability of the actions $a^t$ that occurred during this trial. Even in this case, the plasticity rule is local both in space and time, allowing for an online learning for the agent-network. All the weight updates are implemented using Adam Kingma & Ba (2014) with default parameters and learning rate 0.001.

## 2.4 RESOURCES

The code to run the experiments is written in Python 3. Simulations were executed on a dual-socket server with eight-core Intel(R) Xeon(R) E5-2620 v4 CPU per socket. The cores are clocked at 2.10GHz with HyperThreading enabled, so that each core can run 2 processes, for a total of 32 processes per server. To reproduce each training curve the computation time is $1 - 2 hours$ per realizations (10 realizations).

## 3 RESULTS

We considered a classical benchmark task for learning intelligent behavior from rewards Mnih et al. (2016): winning Atari video games provided by the OpenAI Gym Brockman et al. (2016) (The MIT

License). We considered the case of a limited temporal horizon, $T = 100$. In this case, in the best scenario the agent scores 1 point and in the worst one the opponent scores 2 points. The world variables $\xi_k^t$ represent the paddles and ball coordinates. To win such a game, the agent needs to infer the value of specific actions even if rewards are obtained in a distant future.

Indeed, Atari games still require a large amount of computational resources and interaction with the environment to be solved. Only recently, model-based learning have shown to achieve efficient solutions for such tasks, see e.g. Ye et al. (2021). This is why we introduce the model-based component to outperform the state of the art of reinforcement learning in recurrent spiking networks Bellec et al. (2020).

## 3.1 AWAKE

We defined an "awake" phase (see Fig.1) in which our network interacts with the environment and plays a match. The dynamics of the agents and of the environment are defined by the following equations:

$$
\begin{cases}
\pi_k^{t+1} = \sum_i R_{ki}^A s_{A,i}^{t+1} \\
\xi_k^{t+1} = G(a_k^{t+1}, \xi_k^t) \\
r^{t+1} = F(a_k^{t+1}, \xi_k^t)
\end{cases}
\tag{9}
$$

where the functions $G(a_k^{t+1}, \xi_k^t)$, $F(a_k^{t+1}, \xi_k^t)$ are evaluated by the Atari environment. In this phase, the agents optimizes online its policy, updating its weights in agreement with eq.8. At the same time, the model-network, learns online to predict the effects of agent's actions on the environment (reward and world update) following eq.5.

## 3.2 DREAMING

At the end of this phase, it follows the "dreaming" phase, in which the network is disconnected from the environment. The agent plays a match in its own mind (for a temporal horizon of $T = 50$), interacting with the model-network which replaces the world. In this phase, the agent updates its policy following the same update it would have in the awake phase (see eq.8), while the model is not updated. The initial condition of the variables describing the world $\xi_k^t, r^t$ are extracted randomly, and their dynamics is defined by the equations:

$$
\begin{cases}
\pi_k^{t+1} = \sum_i R_{ki}^A s_{A,i}^{t+1} \\
\xi_k^t = \tilde{G}(a_k^{t+1}, \xi_k^t) = \sum_i \mathsf{R}_{ki}^\xi \bar{s}_{M,i}^t \\
r^t = \tilde{F}(a_k^{t+1}, \xi_k^t) = \sum_i \mathsf{R}_i^r \bar{s}_{M,i}^t
\end{cases}
\tag{10}
$$

where now the functions $\tilde{G}(a_k^{t+1}, \xi_k^t)$ and $\tilde{F}(a_k^{t+1}, \xi_k^t)$ are estimated by the model-network. In this way, it is possible to simulate a Pong game (see Fig.2A for an example of simulated consecutive frames) in which the agent is able to test its policy and to improve it, even when the environment is not accessible. In Fig.2B(left) it is reported an example of the spiking activity in the two network (agent and model top and bottom respectively). In Fig.2B(right) it is shown an example of the output of the two networks during one single dream. It is interesting to see that the change of the paddle position estimated by the model-network, follows the policy computed by the agent-network (e.g., the y coordinate of the paddle increases when the probability to go up is high, green line).

We run an experiment in which we alternate one game in the environment and one dream (see algorithm 1). We report in Fig.2C (orange lines) the performances of the network as the average total reward as a function of the number of interactions with the environment (dashed line, solid line and shading are the average, the 80th percentile and the standard error respectively, evaluated over 10 independent realizations of the experiment). We compare such performances with the ones obtained in the case in which the model does not dream between one (real) game and the other (see Fig.2C black lines). We stress that this condition is equivalent to the approach used in Bellec et al. (2020), the state of the art for reinforcement learning in recurrent spiking networks, that is shown to achieve performances comparable to A3C Mnih et al. (2016). We observe that our model allows

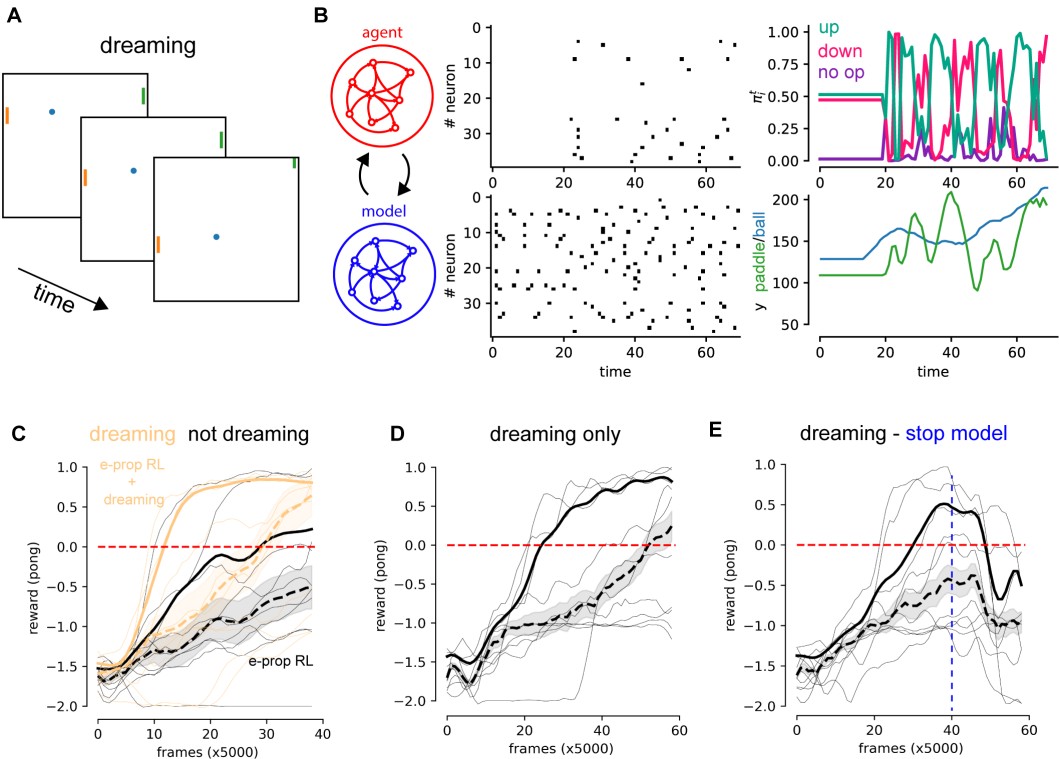

Figure 2: Dreaming. (A) Three consecutive frames of the reconstructed environment during the dreaming phase. (B) Example of the spiking activity in the two sub-networks during a dream (left). (right) Example, of the read out of the two sub-networks, representing the policy and the predicted position of the y paddle. (C) Average (dashed line), standard error (shading), and 80th percentile (solid line) over 10 independent realizations of the achieved reward. Reward as a function of the number of interactions with the environment, with (orange) and without (black) the dreaming phase. Thin lines represent the single realizations. (D) Average (dashed line), standard error (shading) and 80th percentile (solid line) of the reward when the policy gradient is active only during sleep. (E) Same as in (D), but the update of the model is interrupted after 40 (x5000) interactions with the environment (blue dashed vertical line).

for a sensible improvement of the learning speed, significantly reducing the number of interactions with the environment required to achieve desirable performances.

To prove that the model is actually learning during the dreaming phase, we made an experiment in which the policy gradient plasticity rule is active only during the dreaming phase (see Fig.2D), showing the capability to successfully learn to win the game.

However, we note the importance to regularly to compare the model with the reality and to update it. Otherwise, the agent-network cannot improve indefinitely its policy in the simulated environment. We show that when we stop the update of the model, the policy doesn't improve anymore (see Fig.2E, vertical blue line). This is probably due to the fact that after a policy improvement, the agent has access to new states configurations of the environment. E.g., at the beginning it scores few points, and it rarely observes positive rewards. As a consequence, the model-network is not capable to predict positive rewards when it starts scoring more points.

Another contribution to this point, is online learning. This is a choice we made as a constraint for biological plausibility. However, doing so, the model-network tends to forget situations that are not observed since a long time. As an example, if the agent-network is always scoring points, the model-network might forget situations in which a negative reward is received.

---

**Algorithm 1** Dreaming

---
**while** $iteration < N_{iter}$ **do**
    **Awake phase**
    play one game $T = 100$ in the real env.
    update parameters of the agent-network $\{w_{ij}^A, R_{ki}^A\}$            $\triangleright$ policy gradient
    update parameters of the model-network $\{w_{ij}^M, R_{ki}^\xi, R_i^r\}$        $\triangleright$ supervised
    **Dreaming phase**
    play one game $T = 50$ in the simulated env.
    update parameters of the agent-network $\{w_{ij}^A, R_{ki}^A\}$            $\triangleright$ policy gradient
**end while**

---

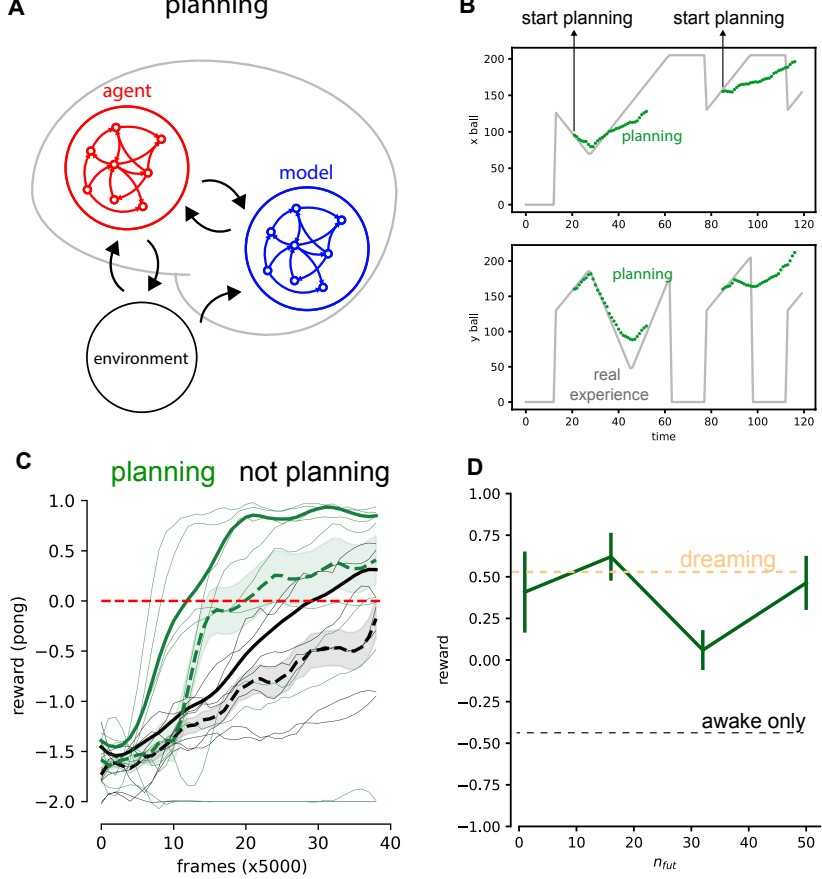

Figure 3: Planning. (A) Sketch of the planning phase, the environment is always available. (B) Example of online "planning", during the observation of the world (gray lines) the model is used to predict $n_{fut}$ steps in the future. (C) Average (dashed line), standard error (shading) and 80th percentile (solid line) of the reward as a function of the number of interactions with the environment, with (green) and without planning (black). (D) Average final reward as a function $n_{fut}$.

### 3.3 PLANNING

We considered an online alternative use of the world model, that is "planning". Contrarily to the "dreaming", this is not a separate phase, but acts in the awake phase (see Fig.3A). There are many possible ways to use planning in reinforcement learning Wang et al. (2019). In this work, we use it, similarly to the dreaming, to increase the data to perform the policy gradient.

In principle, this approach should be much more accurate than "dreaming", since dreams produce long simulated games, possibly producing a large compounding-error. On the contrary, "planning" simulates a few steps $n_{fut}$ (see Fig.3B) in the future every $\Delta t_{pred} = 2n_{fut}$ time steps in the real world (in order to have comparable data augmentation in planning and dreaming), minimizing the effect of the compounding-error. However, "planning" requires online computational resources, while "dreaming" doesn't, since the extra computational cost is required offline, when the world would not be available anyway. For this reason, we report the performance obtained with and without planning for $n_{fut} = 1$(see Fig.3C, respectively green and black). In Fig.3D we show the average reward at the end of the training (average over the last 250 games, and over 10 independent realizations), as a function of $n_{fut}$. In Fig.3D we observe that the performances obtained with dreaming and planning are comparable. This would be enough to prefer dreaming to planning, however it is possible that in other conditions (more complex games, longer temporal horizon) planning would show visible advantages.

Of course the simulation of short sequences, starting from real-world initial conditions, could be done offline, but that would require the storage of detailed information (actions, states, rewards) about the "awake" experience, to define the initial condition of such sequences. This is something that is not biologically plausible, and that cannot be done in a one-shot learning fashion with current learning rules. For these reasons, we discard this approach.

## 3.4 GENERALIZABILITY OF THE APPROACH

### 3.4.1 PONG FROM PIXELS

The task described above is a low-dimensional task from the relevant variables of the pong environments (ball and paddles positions). To show that our framework can be extended to other conditions, we run an additional experiment. We evaluated the Atari Pong from pixels. In order to do so, we opted for a simple approach: we took the black/white frames of the pong game, and randomly projected all the 33600 pixels ($x_i^t$, $i = 1, ...33600$) into a small number of variables ($D = 4$ in our case). As a result, the world variables $\xi_k^{t+1}$ (as described above) are a random combination of the 33600 pixels, $\xi_k^{t+1} = \sum_{h=1}^{33600} F_{kh}x_h$, where $F_{kh}$ are Gaussian weights with zero mean and $0.1$ variance, and $k = 1, ...D$. In other words, we randomly defined a low dimensional latent space. We observed results comparable to what shown in Fig.2c. The approach described above allows, in principle, to apply our method to any other Atari games from pixels.

### 3.4.2 BOXING FROM PIXELS

**When considering Boxing from pixels, we observe that dreaming and planning does not improve agent performances in the random latent space, as it is defined in the previous section. For this reason we used a simple autoencoder to define the world variables, such that $\xi_k^{t+1} = G(\vec{\xi}^t)$ (where $G$ represents the encoder part of the autoencoder, and $\vec{\xi}$ a vector whose components are $\xi_k^{t+1}$). The decoder is only used for visual purposes, to reconstruct the predicted future frames (see Fig.4B, bottom). The autoencoder is simply defined as a 3 (encoder) + 3 (decoder) layer (33600,128,64,36,64,128, 33600) feed forward network, trained on $10000$ frames collected from the environment using a random policy. In this case, the dimensionality of the latent space describing the world was $D = 36$. We show that planning, in this latent space, shows performances advantages (see Fig.4A).**

## 4 CONCLUSION

We presented a two-module spiking network, capable of performing two alternative versions of model-based biologically plausible reinforcement learning: "dreaming" and "planning". We demonstrate that the simulated world observation produced by the model-network, provide valuable data for the agent-network to improve its policy, significantly reducing the necessary number of interactions with the real environment. A major goal of our model, is to achieve a high level of biological plausibility. For this reason, we propose plasticity learning rules that are local in space and time. This is a major requirement in biology, since synapses have only access to local information (e.g., the activities of the pre- and post-synaptic neuron and the reward at that time). Moreover, this al-

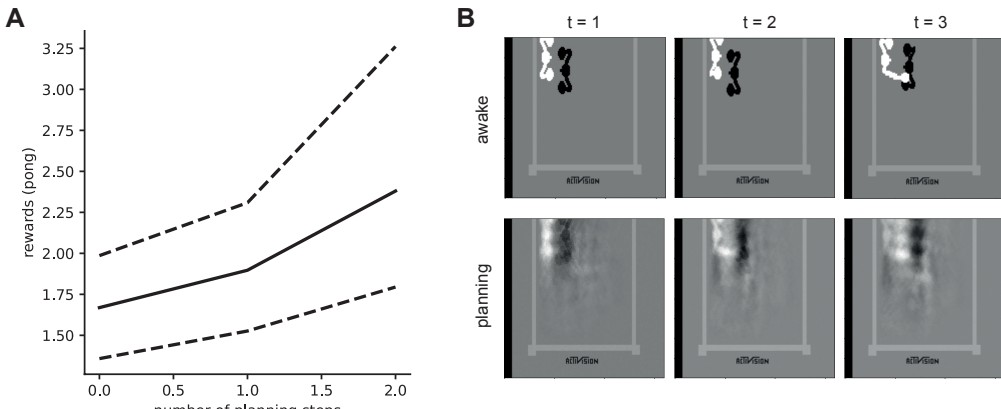

Figure 4: (A) Average reward over 10 realizations, as a function of number of planning steps. (B) Top. Three consecutive frames in the environment. Bottom. Three consecutive reconstruced predicted frames.

lows for an online training of our networks. Usually, training the world-model requires the storage of an agent's experiences Ye et al. (2021); Abbeel et al. (2006); Schrittwieser et al. (2020); Ha & Schmidhuber (2018); Kaiser et al. (2019); Hafner et al. (2020) to offline learn the model. Besides being implausible, storage of long sequences is highly problematic on neuromorphic and FPGA platforms, where memory resources are scarce, and are in general dedicated to the storage of the synapses describing the model. The storage of a temporal sequence would require an order $O(T)$ memory (where T is the number of temporal steps), hence saturating the available memory. A typical solution is to store externally the temporal sequences, but that would imply a latency of order $O(T)$ since the available bandwidth for memory access is much lower than the one used to exchange signals internal to the network. In our case, the possibility to learn online, makes it unnecessary to store any information on the agent's experience. We observe an interesting tradeoff between "dreaming" and "planning". Dreaming, simulates long sequences in the model-based simulated environment. Usually, this approach is not efficient because of the large compounding errors, but it can be comfortably performed offline. On the other hand, planning simulates shorter sequences, but requires computation online, while the network is already performing the task. The comparability of performances for planning and dreaming (for the same amount of simulated generated data) underlines the importance of the "dreaming" learning phase in biological and artificial intelligence. Our model provides a proof of concept that even small and cheap networks with online learning rule can learn and exploit world models to boost learning. Our work is a step forward, towards building efficient neuromorphic systems for autonomous robots, to efficiently learn in a real world environment. "Dreaming" intervenes when the robot is no longer able to explore the environment (it is not accessible anymore, the robot is low energy). In this condition, the robot optimizes learning by reasoning in its own mind. These approaches are of great relevance when the acquisition from the environment is slow, expensive (robotics) or unsafe (autonomous driving).

## 4.1 LIMITATIONS OF THE STUDY

This study, shows a promising biologically plausible implementation of efficient model-based reinforcement learning, proving that it is possible to speed up learning in biological-based networks. However, we considered a specific task with a limited temporal horizon (T = 100). We plan to test our approach on more complex and long-horizon tasks. **Even though we are capable to show the effectiveness of our model in different task (atari Pong and Boxing) and conditions (from RAM and from pixels), it remains crucial to look at how the performance scales with the size of the network and the dimensionality of the problem. A possible generalized task can take inspiration from what is used in human experiments with multiple cue learning tasks Osman (2010).That would be a good testbed to see how the performance scales with the complexity of the task, and how they compare to human performances.**

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
