# OpenReview forum: "Towards biologically plausible Dreaming and Planning"
_ICLR.cc/2023/Conference — Submitted to ICLR 2023_

### Official Review · Reviewer_w9LE · 2022-10-23

**Confidence:** 3
**Correctness:** 2
**Technical Novelty And Significance:** 2
**Empirical Novelty And Significance:** 1
**Recommendation:** 1

**Clarity, Quality, Novelty And Reproducibility:**

There are several concerns regarding the clarity of the paper.

1. In Section 3.4, the authors mentioned 33600 pixels. Could you please explain why 33600 pixels? The most common setting might be 84 x 84 x 4 (stack) x 3 (RGB) = 84672 pixels (or 28224 with gray image).

2. In Section 2.1, h in $w_{ih}^{A,in}$ is not defined and in Section 2.2, $R_{ki}^\xi$ is not defined.

3. In Figures 2 and 3, the 80th percentile makes confusing. I recommend removing the dashed line.

4. How to measure the “compounding-error” in Section 3.3? Is it related to Figure 3. B?

5. The range of the reward of Pong is [-22, 22]. Is it normalized to -2 to 1?


**Strength And Weaknesses:**

**[Strength]**

1. The idea of applying dreaming-awake in reinforcement learning is interesting.

**[Weakness]**

1. In Section 2.3, the authors mention supplementary material for derivation, but it is not submitted.

2. The authors only conduct experiments on Pong which is one of the most simple games on Atari. Although the authors argue that the proposed method is applicable to any Atari games, the actual experiment should be conducted.

3. In Section 4.1, what is the difference between an image-based task and a state-based task? Do the authors use the ROM version of the Atari environment, not the image version?

4. It seems that it is impossible to use dreaming and planning at the same time.

5. In Figure 1.D, how the background and the ball shape are changed during the dreaming phase?

6. The authors should compare with [Ref 1].

**[minor]**

1. Please differentiate \citep and \citet. It seems that all citations are \citet.

2. Typo:

1) In Section 3.2, all Fig.1 should be changed to Fig.2

2) In Section 1, “Okada & Taniguchi(2021) the authors” => Okada & Taniguchi(2021), the authors”

3) In Section 2.2, “membrane potential.The” => “membrane potential. The”

4) In Section 3.3, since dreams produces => since dreams produce.

[Ref 1] Ellis, Kevin, et al. "Dreamcoder: Growing generalizable, interpretable knowledge with wake-sleep bayesian program learning." PLDI. 2021.



**Summary Of The Paper:**

The paper proposed two strategies for biologically plausible learning rules; dreaming and planning. In Dreaming-Awake, the agent builds the world model. In the awake phase, the agent interacts with the environment, and in the dreaming phase, the agent interacts with the world model to update the weight. In the “planning” strategy, the agent interacts with both the world model and the environment.


**Summary Of The Review:**

In general, although the idea of dreaming, awake, and planning is interesting, the experiment is only conducted in Pong and has many questions about results and settings.

-- Post Discussion Period --

After reading the authors' responses and other reviews, I maintain my rating. Although some concerns are resolved, there is still an unsolved concern and further analysis is needed including other environments, not just boxing and pong.

---

> ### Author Response · Authors · 2022-11-18
> **Some clarification**
>
> Weakness
>
> 1. We apologize, supplementary materials are indeed not present, we refer to [2] for derivation.
> 2. We run an additional experiment, we tested our model on another atari game: Boxing from pixels. The results are included in a new Section 3.4.2.
> 3. With state-based refers to a model that operates in the space of internal states encoding the observation of the world (e.g. mu zero [3]). However, we acknowledge that this sentence is unclear, and we removed it.
> 4. It is actually possible to use both of them. Planning operates online at every time-step. Dreaming is active offline, at the end of the experience.
> 5. In this case, Pong is played observing the state of the RAM, and we have direct access to the position of the ball and paddles. The frame during dreaming is reconstructed by placing a point (the ball) and the segments (the paddles) in the positions predicted by the model network.
> 6. We included this reference in the paper.
>
> Clarity
>
> 1. We do not stack, use gray scale, and consider the original resolution (not down-scaled). This results in 33600 pixels.
> 2. h is the presynaptic index. We included this sentence at the beginning of the section: “synaptic matrix $w_{ij}$ (from neuron $j$ to neuron $i$)”.
> 3. We like to show the average performances, and the realizations with better performances.
> 4. In model-based reinforcement learning, the model can be composed with itself to enable predicting multiple steps into the future, but one-step prediction errors can get magnified, leading to unacceptable inaccuracy. This is usually referred to as compounding-error.
> 5. The temporal horizon is 100 frames. In this temporal interval, the maximum score is 1 and the minimum is -2.
>
> [Ref 1] Ellis, Kevin, et al. "Dreamcoder: Growing generalizable, interpretable knowledge with wake-sleep bayesian program learning." PLDI. 2021.
> [2] Bellec, G., Scherr, F., Subramoney, A., Hajek, E., Salaj, D., Legenstein, R., & Maass, W. (2020). A solution to the learning dilemma for recurrent networks of spiking neurons. Nature communications, 11(1), 1-15.
> [3] Schrittwieser, J., Antonoglou, I., Hubert, T. et al. Mastering Atari, Go, chess and shogi by planning with a learned model. Nature 588, 604–609 (2020). https://doi.org/10.1038/s41586-020-03051-4

---

> > ### Comment · Reviewer_w9LE · 2022-11-18
> > **Thank you for response**
> >
> > Thank you for the clarification and revision.
> >
> > > The temporal horizon is 100 frames. In this temporal interval, the maximum score is 1 and the minimum is -2.
> >
> > It seems that it is trained on 20k (40 x 5000) frames. Could you elaborate on this?
> >
> > typo:
> > On Page 9, Osman(2010).That -> Osman(2010). That

---

> > > ### Author Response · Authors · 2022-11-21
> > > **Frames**
> > >
> > > One frame is one interaction with the environments. In other words, the frames are the number of temporal steps in the environment.
> > > The more the frames, the more are the data observed by the model.
> > > The average reward as a function of frames is usually reported in reinforcement learning to show the performances of the learning algorithm.
> > >
> > > In our case, 20k frames are 200 games, each composed of 100 frames.

---

> > > > ### Comment · Reviewer_w9LE · 2022-11-23
> > > > **Re: Frames**
> > > >
> > > > I appreciate the author's clarification. If I understood correctly, the authors used the same value for the temporal horizon and the maximum episode length (if the episode length is longer than this value, reset the environment). In reinforcement learning literature, people use much longer episode lengths ~1000 or even ~10000 so that the agent sufficiently explores the environment. Could you please elaborate on only focusing on the first 100 frames?

---

### Official Review · Reviewer_k79x · 2022-10-25

**Confidence:** 3
**Correctness:** 3
**Technical Novelty And Significance:** 2
**Empirical Novelty And Significance:** 1
**Recommendation:** 3

**Clarity, Quality, Novelty And Reproducibility:**

The text is well-written including a comprehensive introduction to the model, relevant literature, and the methods. The experiments are performed in accordance with what’s standard in the field of spiking neural networks. This makes the experiments reproducible.

**Strength And Weaknesses:**

The paper introduces an interesting and timely advance to the field of spiking neural networks by extending the existing RL approaches with a model-based part in that domain. The work is technically sound; the theory and the experiments are well-explained.

The weaknesses here are mostly specific not to this work, but rather to the field of spiking neural networks.

-The biological plausibility here is conditioned on the locality of the learning rules. This locality, in turn, is conditioned on the network being composed of one recurrent layer and one linear readout layer. Indeed, a similar locality of the learning rules can be achieved in a similarly structured non-spiking neural network train with backpropagation (if BPTT is not used).

-The task at hand (ATARI Pong, mostly with non-pixel input) is relatively simple, thus it can be solved with a relatively simple neural network. It is unclear whether this approach would scale to more-complex tasks normally requiring deeper networks.

-The paper mentions the potential for deploying the spiking RNN architecture in robots or on neuromorphic chips but for now these applications for this model remain theoretical. In practice, the model was trained on a conventional Python package and took much longer to train than a similar non-spiking model would.

Similar papers accepted to ICLR typically present a finding which could either be readily deployed in practice outperforming the existing approaches or could deepen our theoretical understanding of deep-learning models and/or biological objects. While the current work is interesting, well-conducted, and timely for its field, it may not fall into the scope of what’s usually accepted to ICLR.


**Summary Of The Paper:**

This work introduces model-based reinforcement learning to the domain of spiking recurrent neural networks. To this end, the authors used two subnetworks – an “agent” one for computing policy and a “model” one for predicting future rewards and states of the environment. The authors then formulated local learning rules for their network and tested it on the ATARI Pong game where they observed higher sample efficiency compared to model-free spiking RNN approaches.

**Summary Of The Review:**

The paper presents a well-conducted study on adding model-based functionality to spiking RNN-based RL. Whereas the study is timely and interesting in the field, it may not deliver general enough results which could either extend our understanding of biological systems or be used in practice, so, currently, it may not be a sufficient contribution to the general ICLR community.

---

> ### Author Response · Authors · 2022-11-18
> **The strength of spiking networks**
>
> Learning rule.
> The learning rule we use is e-prop [1] that is a bio-inspired approximation of BPTT. It allows implementing BPTT in a network of spiking neurons, and to update online the weights of the network, making learning extremely more efficient and computationally less expensive.
>
> Complexity of the task.
> We tested our model on another atari game: Boxing from pixels. The results are included in a new Section 3.4.2.
>
> Hardware implementation.
> We acknowledge that our model is a simulated experiment. However, we argue that the use of (1) spiking neurons, (2) online learning rules, (3) memories stored in the network (and not in an external storage), make our model, almost straightforward to be implemented in a neuromorphic hardware. We included such a statement at the end of the introduction.
>
> [1] Bellec, G., Scherr, F., Subramoney, A., Hajek, E., Salaj, D., Legenstein, R., & Maass, W. (2020). A solution to the learning dilemma for recurrent networks of spiking neurons. Nature communications, 11(1), 1-15.

---

> > ### Comment · Reviewer_k79x · 2022-11-19
> > **Re:**
> >
> > Thanks for your response.
> >
> > Although I would like to acknowledge the Authors' effort to address the Reviewers' comments by testing their model on an additional, more complex task (Boxing from pixels), the results here are not compared to any reference values and therefore it's hard to judge whether the use of the proposed model has resulted in any significant improvement compared to the available baseline models. I sure hope that additional research in this direction may help strengthen the paper. For now, though, the scope of the reported results and the description of the new results necessitate keeping the initial valuation of the paper unchanged.

---

> > > ### Author Response · Authors · 2022-11-21
> > > **Model reference**
> > >
> > > As well as in figure 2, we compare our model, with the state of the art for reinforcement learning in recurrent spiking networks, e-prop [1].
> > > As discussed in the paper, e-prop is shown to achieve performances comparable to A3C, a powerful algorithm in the field of reinforcement learning.
> > >
> > > In our new experiment (Boxing from pixels, reported in Fig4) the reward with 0 planning steps is equivalent to using e-prop (this is our reference).
> > > We show that using 1 or 2 planning steps (our model) provides a clear advantages in terms of performances.
> > >
> > >
> > > [1] Bellec, G., Scherr, F., Subramoney, A., Hajek, E., Salaj, D., Legenstein, R., & Maass, W. (2020). A solution to the learning dilemma for recurrent networks of spiking neurons. Nature communications, 11(1), 1-15.

---

> > > > ### Comment · Reviewer_k79x · 2022-11-22
> > > > **Acknowledged**
> > > >
> > > > Thanks for the clarification, I've totally missed it in the added text and the new figure. As this is an exciting result, I think the paper would benefit from it being highlighted in all ways possible, including the explicit mention of the reward with 0 planning steps being equivalent to using e-prop (in the figure) and, perhaps, adding your response here as a paragraph to the main text where you describe the boxing-from-pixels experiment. Would've loved to bump up the score to 4 but the system only allows odd numbers, which is a bit too much.

---

### Official Review · Reviewer_MjQE · 2022-10-25

**Confidence:** 3
**Correctness:** 3
**Technical Novelty And Significance:** 3
**Empirical Novelty And Significance:** 3
**Recommendation:** 6

**Clarity, Quality, Novelty And Reproducibility:**

The paper is generally well-written and easy to follow.
A few minor comments:
- Fig 1 is referred to instead of Fig 2 in a couple of lines on pages 6 and 7
- The caption for fig 3B refers to orange lines; it should be gray instead?

**Strength And Weaknesses:**

**Strengths**:
- This work addresses the challenge of developing an efficient, biologically plausible model based RL method. This is of high relevance for applications such as building efficient neuromorphic systems for autonomous robots, autonomous driving, etc.
- Local learning rules are derived than enable online training of the recurrent spiking networks
- A concrete instantiation of dreaming is provided, and its benefit in learning is clearly demonstrated

**Weaknesses**:
The authors clearly state the limitations of their study:
- The approach is tested only on a simple game with direct access to states and a short time horizon
Nevertheless, it is a concrete step towards achieving biologically plausible and efficient model-based reinforcement learning.


**Summary Of The Paper:**

This work presents a biologically plausible model-based RL approach that uses dreaming and planning to to efficiently use the learnt world model. The model comprises two recurrent spiking network modules, (i) to compute the policy to behave in an environment, and (ii) to learn to predict the next state of the environment given the past state and the action. The authors derive biologically plausible learning rules (local in both space and time) to train these modules. In the awake phase, the agent optimizes its policy and learns to predict the effects of its actions on the environment (i.e., both spiking network modules are updated). In the dream phase, however, the agent does not have access to the environment. Instead it uses its current model of the world to simulate experiences and uses these to optimize its policy. Planning is another approach used while the agent is in the awake phase to increase the data use to optimize the policy. Finally, the two-module spiking network is trained on a simplified version of the Atari pong game and the roles of dreaming and planning are evaluated.

**Summary Of The Review:**

The paper addresses the important problem of achieving biologically plausible and efficient model-based reinforcement learning. While the proposed network is evaluated only on a very simple task, it is nevertheless a concrete step towards achieving this goal. Overall, the paper is well presented and the proposed model is promising and could potentially lead to interesting avenues of future work.

---

> ### Author Response · Authors · 2022-11-18
> **Generalizability of the approach**
>
> We acknowledge that evaluation is the major weakness of our paper. For this reason, we tested our model on another atari game: Boxing from pixels. The results are included in a new Section 3.4.2.

---

### Official Review · Reviewer_7i2g · 2022-10-27

**Confidence:** 3
**Correctness:** 2
**Technical Novelty And Significance:** 3
**Empirical Novelty And Significance:** 2
**Recommendation:** 3

**Clarity, Quality, Novelty And Reproducibility:**

** Clarity **

The paper is generally well written and is a pleasure to read.

I feel that the fact that the paper has many ambitious goals at once sometimes makes the paper harder to follow. For example, neither the title nor the abstract mention spiking networks, and create the impression that the idea of "dreaming" is the key invention of the paper. At the same time, arguably, it's the tricky adaptation of spiking networks to the task which is the main achievement of the paper.

This and similar issues complicate reading and result in an impression that the paper attempts to balance a number of related, but not equivalent goals.

** Quality **

My main issue with the experimental planning is that the authors list many ambitious goals, but none of them (in my opinion) receives commensurate depth of exploration and justification. For example, consider the following paragraph:

"To our knowledge, there are no previous works proposing biologically plausible model-based reinforcement learning in recurrent spiking networks. Our work is a step toward building efficient neuromorphic systems for autonomous robots, capable of learning new skills in real-world environments. Even when the environment is no longer accessible, the robot optimizes learning by reasoning in its own mind. These approaches are of great relevance when the acquisition from the environment is slow, expensive (robotics) or unsafe (autonomous driving)."

While I agree that using spiking networks is indeed more biologically plausible than traditional artificial neural networks, I found the discussion of biological plausibility insufficiently deep. For example, the authors stress the fact that memory replay is biologically implausible, but I can not fully agree with that. Humans have reasonable episodic memory. Even though it is not infinite capacity, I don't believe that one can discard all episodic replay as biologically implausible. A deeper look is warranted into how the effects of dreaming in spiking networks is more similar to human dreaming than using, say, exponentially decaying replay buffers in traditional reinforcement learning algorithms.

Additionally, since learning is not done online, the biological plausibility argument becomes even less convincing.

The claims about the usefulness of the approach in autonomous driving and robotics similarly don't receive enough confirmation. While I understand the difficulties associated with training spiking networks and I don't expect a state-of-the-art performance across a range of image-based RL tasks, it is still crucial to look at how the performance scales with the size of the network and the dimensionality of the problem. For example, a toy task, similar to what is used in human experiments with multiple cue learning tasks (see https://psycnet.apa.org/record/2009-24669-008 for a review) could be a good testbed to see how the performance scales with the complexity of the task (It could also show that, potentially, spiking neural networks produce more human-like behaviour, and be used to investigate biological plausibility).

Lastly, the main limitation of all model-based approaches is that what happens if the model can not easily fit the environment. In humans, we know that dreams do not copy our real experience, but are still apparently useful in learning. If the "dreaming" part of the contribution is key, a deeper look into what happens when the model can not properly fit the task is warranted. Additionally, perhaps a more systematic investigation (on simulated tasks) is warranted, to answer the question of what are the conditions when dreaming is useful. For example, are there task where even an important model is very useful? Are there cases where learning a model is easier/faster than learning a policy? And so on.

** Novelty/Impact **

While the approach is sufficiently novel, the novelty of the paper is not high enough to be its main selling point. The work is heavily building upon existing ideas, which puts more importance on the depth and thoroughness of experimental support.

** Reproducibility **

The approach is described clearly, but the code is not provided. I can imagine that some researchers might struggle with implementing the algorithms in an exactly the same way as the authors and reproducing the results.

** Typos/phrasing **

still, a clear and coherent understanding of the mechanisms that induce generalized beneficial effects is still missing - "still" is repeated.

Taking inspiration from biology, an intriguing idea is that one possible usage of a learned model, is during periods in which the neural network is offline. -- I'd suggest rephrasing. E.g. "Taking inspiration from biology, we explore an intriguing idea that a learned model can be used when the neural network is offline."

**Strength And Weaknesses:**

Strengths:
- The paper addresses a number of highly relevant and important problems
- Using spiking networks for reinforcement learning is fundamentally challenging, but the authors manage to successfully handle the task
- The authors consider an expansion of their method to image-based inputs

Weaknesses:
- The paper focuses on many targets at once, such as biological plausibility, applicability in robotics, suitability to neuromophic hardware, as well as general usefulness of the approach as a general-purpose reinforcement learning algorithm. I believe that while some of these goals are correlated, they are not the same. As a result, the paper produces an unfocused impression. I believe that narrowing down the range of claims and delving deeper into one of these topics would greatly benefit the paper.
- The range of applications considered is limited. The method works reasonably well when applied to problems that are fundamentally extremely low-dimensional, but it's not clear whether more complex systems can be managed in a similar manner. While the authors say that "The approach described above allows, in principle, to apply our method to any other Atari games from pixels", it remains unclear whether this suggestion is practical.
-  The novelty of the approach is limited. While the use of spiking neural networks in this specific context is novel, the idea of using model-based simulations or dreaming to enhance learning is very well known. Again, I believe that additional, more focused results might help to mitigate the issue.

I clarify and expand upon some of these points in the main body of my review.

**Summary Of The Paper:**

The authors propose a recurrent spiking network-based reinforcement learning algorithm, in which the network consists of two major parts (an agent and a model). The agent is trained to act using policy gradient, while the model is trained to predict the effect's of the agent's actions (both the reward and the state transition). The authors apply their approach to the game of Pong, showing that dreaming indeed improves performance. They also consider an alternative to dreaming (namely, planning), which showed similar performance.

**Summary Of The Review:**

Overall, the paper presents a number of fascinating ideas and has a huge potential. At present, unfortunately, I believe that the balance between breadth and depth was not optimal, which does not allow me to recommend the paper for acceptance.

I believe that re-structuring the paper, either around its main technical achievement - adapting spiking networks to work in new circumstances, or around its main conceptual claim (biological plausibility) would greatly benefit the paper. In both cases, however, additional experiments would be needed to increase the impact of the contribution, and to more fully understand the properties of the proposed system.

** UPD **

After reading other reviews and the authors' responses, I leave my assessment unchanged. While some concerns were addressed, I believe that a more thorough reworking of the paper is necessary for it to be up to the standards of the ICLR conference.

---

> ### Author Response · Authors · 2022-11-18
> **Narrowing the scope and plausibility**
>
> We thank the reviewer for the precious comments. In particular, we understand the confusion that might be created in the different goals suggested in the papers.
>
> Our main focus is to build a spiking network capable to learn and replay with a bio-inspired architecture, in order to be easily ported to hardware  (e.g. neuromorphic). Importantly, this approach makes unnecessary the use of external storage, which is integrated within the network, allowing to avoid the associated latencies, and making learning more efficient. For this reason, we rewrote the last paragraph of the introduction to make it narrower.
>
> About the plausibility of experience replay, we agree with the observation of the reviewer, and we included the following sentence in the text: <<Indeed, even though the brain is capable to store episodic memories, it is implausible that it is capable to sample offline from ten or hundreds of past experiences.>>
>
> Moreover, we remark that the learning is performed online, both for agent and model networks. This means that weights are updated at every time-step, using variables that are accessible locally in space and time, without storing any information.
>
> In order to provide more experimental support, and understand how our approach scales with the complexity, we tested our model on another atari game: Boxing from pixels. The results are included in a new Section 3.4.2.
>
> We appreciate the suggestion about tasks inspired by experiments with humans, but we did not have time to implement them yet. However, we discuss this point in the “limitations of the study”.

---

> > ### Comment · Reviewer_7i2g · 2022-11-23
> > **Thank you for your response**
> >
> > I appreciate your thoughtful response. Unfortunately, I think that the changes necessary are a bit too global to be addressed in the limited timeframe available, as I believe that the paper needs to be a bit more substantially rewritten and re-focused. For this reason, I can not quite push my evaluation higher, but I do think that the paper has great potential and I would love to see it's revised version published one day!

---

### Decision · Program_Chairs · 2023-01-20

**Decision:**

Reject

**Justification For Why Not Higher Score:**

This paper does not provide any new theoretical insights, nor any clear empirical advances over previous approaches. It is at best a limited increase in biological plausibility of these models. I do not think this is enough for acceptance at ICLR.

**Justification For Why Not Lower Score:**

N/A

**Metareview: Summary, Strengths And Weaknesses:**

This paper presents a spiking neural network model for reinforcement learning that utilizes two distinct modules, one an actor, and the other a model, which is used to learn transition dynamics for training the actor and planning. The authors show that the network can learn to play Pong and Boxing more rapidly than a network without the model module.

The reviewers were concerned about the limited tests provided, and precisely what the contributions of this paper are, given that there are other models that do improved RL via dreaming/planning, but at higher levels of performance and with more rigorous testing (e.g. https://arxiv.org/abs/2010.02193).

Based on the scores alone (average of 3.25) this is a fairly clear reject case. Moreover, my judgement as an AC is that the contributions of this paper are extremely limited, and can largely be summed up as seeing whether an approach that others have taken before can work with spiking networks trained with e-prop. In my assessment, this is an insufficiently important contribution to warrant publication at ICLR.

**Summary Of Ac-Reviewer Meeting:**

N/A